# DeepSDCS: Dissecting cancer proliferation heterogeneity in Ki67 digital whole slide images

**Priya Lakshmi Narayanan**
Centre for Evolution and Cancer,
Division of Molecular Pathology,
Institute of Cancer Research
London, United Kingdom
priya.narayanan@icr.ac.uk

**Shan E Ahmed Raza**
Centre for Evolution and Cancer,
Division of Molecular Pathology,
Institute of Cancer Research
London, United Kingdom
shan.raza@icr.ac.uk

**Andrew Dodson**
Academic Department of Biochemistry
Royal Marsden Hospital
London, United Kingdom
andrew.dodson@icr.ac.uk

**Barry Gusterson**
Centre for Evolution and Cancer
Institute of Cancer Research
London, United Kingdom
barrygusterson@icr.ac.uk

**Mitchell Dowsett**
The Breast Cancer Now Toby Robins Research Centre,
Institute of Cancer Research
Academic Department of Biochemistry,
Royal Marsden Hospital
London, United Kingdom
mitchell.dowsett@icr.ac.uk

**Yinyin Yuan**
Centre for Evolution and Cancer,
Division of Molecular Pathology,
Institute of Cancer Research
London, United Kingdom
yinyin.yuan@icr.ac.uk

## Abstract

Ki67 is an important biomarker for breast cancer. Classification of positive and negative Ki67 cells in histology slides is a common approach to determine cancer proliferation status. However, there is a lack of generalizable and accurate methods to automate Ki67 scoring in large-scale patient cohorts. In this work, we have employed a novel deep learning technique based on hypercolumn descriptors for cell classification in Ki67 images. Specifically, we developed the Simultaneous Detection and Cell Segmentation (DeepSDCS) network to perform cell segmentation and detection. VGG16 network was used for the training and fine tuning to training data. We extracted the hypercolumn descriptors of each cell to form the vector of activation from specific layers to capture features at different granularity. Features from these layers that correspond to the same pixel were propagated using a stochastic gradient descent optimizer to yield the detection of the nuclei and the final cell segmentations. Subsequently, seeds generated from cell segmentation were propagated to a spatially constrained convolutional neural network for the classification of the cells into stromal, lymphocyte, Ki67-positive cancer cell, and Ki67-negative cancer cell. We validated its accuracy in the context of a large-scale clinical trial of oestrogen-receptor-positive breast cancer. We achieved 99.06% and 89.59% accuracy on two separate test sets of Ki67 stained breast cancer dataset comprising biopsy and whole-slide images.

1st Conference on Medical Imaging with Deep Learning (MIDL 2018), Amsterdam, The Netherlands.

# 1 Introduction

Recent evidence has highlighted that breast cancer is a biologically, clinically, and molecularly heterogeneous entity [1]. Uncontrolled proliferation is one of the key hallmarks of cancer [2]. Proliferation is often measured by using the Ki67 biomarker in breast tissue images. Ki67 is a nuclear protein expressed in all active phases (G1, S, G2 and M) of the cell cycle [3]. Cell proliferation is controlled by regulatory proteins and the complex tumor environment transcends through the checkpoints of the cell cycle. Ki67 has been validated as a biomarker for evaluating clinical benefits from endocrine treatment. Percentage of Ki67 positivity has been shown to be associated with patient prognosis [4]. However, manual estimation of Ki67 proliferation index in breast carcinoma can be laborious and prone to intra- and inter observer variations [5-7].

Therefore, there is an urgent need to build robust image analysis pipeline and offer standardized diagnostic solution for Ki67 immunohistochemistry (IHC) assay. Previous work in GeparTrio study [8] used Ki67 automated scoring and its validation was selectively based on few regions. Usage of only few regions for validation restricts the complete heterogeneity based challenges underlying in the Ki67 images. A software platform (QuPath) was developed to process TMA images of Ki67, ER, PR, HER2 and p53 expression [9]. This study uses classical texture feature and the machine learning pipeline that could be interactively trained and adapted to the dataset. However, diagnostic slides from clinical trials are highly heterogeneous, posing challenges such as consistent tissue identification across needle biopsy and whole-slide images, intra-class variability of cells and staining variability. Thus, it is critical to develop improved, automated and efficient ways to identify Ki67-positive and negative nuclei on diagnostic Ki67 images.

Recent developments in deep learning, in the context of histology image processing for detection of mitosis in hematoxylin and eosin (H&E) images based on convolution neural network (CNN) was the earliest implementation [10]. In this approach, the CNN network was trained to regress for probability of each pixel extracted from the patch. For cell classification, spatially constrained CNN has been proposed by adding a constrained layer to the network and regressing the final probability map to find the local maxima [11]

In the context of nucleus detection in Ki67 images, a major challenge is the uneven chromatin distribution of hematoxylin stain that frequently occurs in clinical samples. The sharp contrast between heterogeneous weak hematoxylin stains and the Ki67-positive cells often results in failed cell detection. To address this, we design a new approach for localization, segmentation and classification of cells based on hypercolumn descriptors. These multi-scale descriptors not only capture the semantics but also aid localization of information across multiple patches [12]. Thus, our main contribution is the development and extraction of hypercolumn descriptors for a given pixel, which integrate activations from multiple convolutional layers of our SDCS network to yield accurate detection of all Ki67-positive and negative nuclei simultaneously in a large image. Specifically, we exploit the semantic and localization features retained in hypercolumns to address cell segmentation and classification challenges and demonstrated its performance in two clinical cohorts.

# 2 Methodology

## 2.1 Overview

Given an input immunohistological image $i$, the problem is to find a set of detections and then classify the detections into Ki67-positive, Ki67 -negative, stroma and lymphocyte. The problem is solved by training the SDCS detection network and propagating it to the constrained network using the same ground truth annotation coordinates used in the training stages of detection and classification.

## 2.2 Manual annotation of training and testing samples

Twenty whole slide images were selected as training images for the study. An expert pathologist annotated the Ki67-positive cancer cells, Ki67-negative cancer cells (henceforth referred to as Ki67-positive and Ki67-negative cells), stromal cells and lymphocytes. We included samples across the spectrum of Ki67 expression positivity in our training set. Figure. 1 shows two exemplary images from a training sample that has low Ki67 expression. Quality control was performed to exclude the

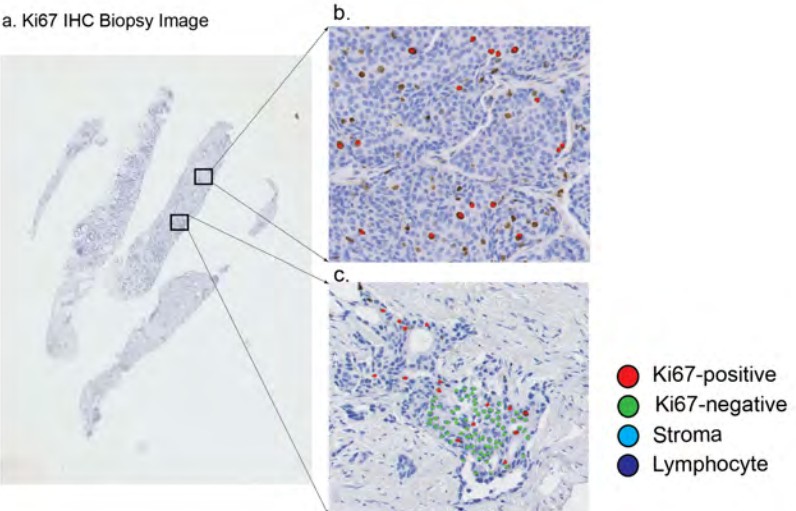

Figure 1: a) An exemplary Ki67 biopsy image, with the annotations for the positive Ki67 nuclei (red), negative Ki67 nuclei (green) marked by the expert pathologist for training our SDCS network. b) Exemplary sparsely scattered positive nuclei showing the proliferation and dense negative nuclei pattern on the biopsy image. c) Tile images (2000x2000) are extracted using openslide library and the annotations are saved in the high resolution 20x tiles. The coordinates are extracted from the base magnification to extract the training patches.

out of focus samples from the training dataset. We used two sets of testing data acquired at different time points to test the performance of cell classification. Breakdown of the number of cells in training validation and test set samples is tabulated in Table 1.

Table 1: Breakdown of the number of annotated cells in training, validation and two testing sets

| Cell types | Training | Validation | Testing set1 | Testing set2 |
|---|---|---|---|---|
| Ki67 Positive | 1626 | 201 | 254 | 386 |
| Ki67 Negative | 733 | 232 | 416 | 744 |
| Stroma | 933 | 267 | 109 | 213 |
| Lymphocyte | 928 | 109 | 256 | 362 |
| Total | 4220 | 809 | 1035 | 1705 |

## 2.3  Hypercolumns

A typical architecture of CNNs consists of convolution layers and max pooling layers followed by a fully connected layer. CNNs use convolution and pooling operations and the pooled image downsamples the input images by the pooling parameter.

During the training phase, we feed the input image to a VGG16 network [13] and extract the sparse hypercolumn descriptors from selected convolutional layers. The hypercolumns are formed by concatenating a series of activations of the convolutional layers. In our implementation, we chose the activations from the final convolutional layer of $c_1$ (p), $c_2$(p) and $c_5$ (p) from the customized VGG16 like architecture (i.e. conv1; conv2; and conv5) to form the hypercolumn. The fully connected layers in the original VGG16 network are implemented as 1x1 convolution layers. As each convolutional block is preceded by a max-pooling operation that downsamples the activations, we perform bilinear upsampling using an appropriate scaling factor such that the resulting resolution for the activations of each layer forming the hypercolumn is 64x64. Then, we sparsely sample random points from the dense hypercolumns to form rich descriptors for a given

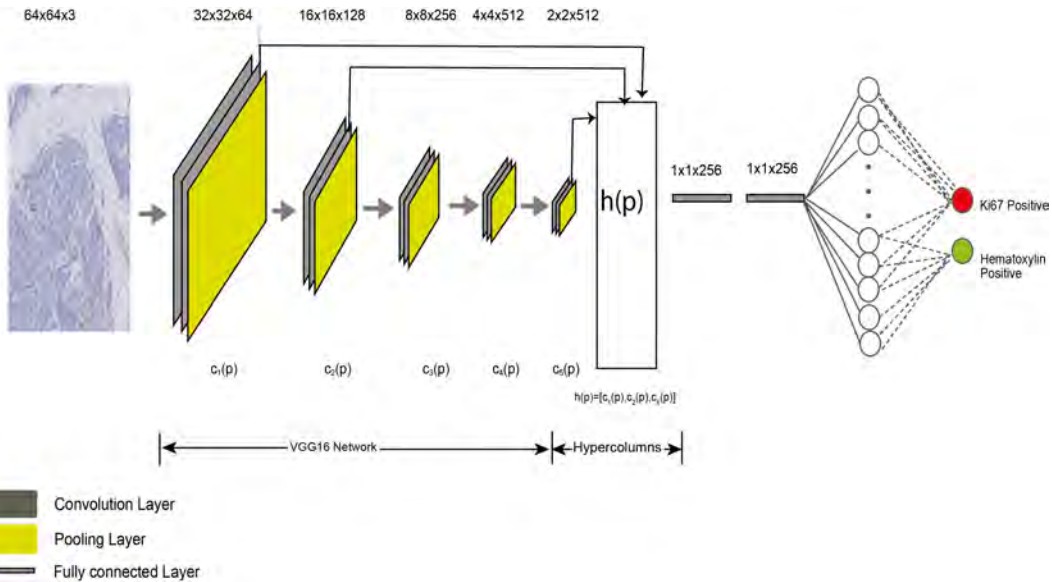

Figure 2: Schematic of our simultaneous detection and cell segmentation (SDCS) network architecture. The patches extracted from the cell annotation locations are used to extract the hypercolumn descriptors. The hypercolumn descriptors are then fed to the dense multi-layer perceptron network to segment them into Ki67 positive and Ki67 negative cells.

pixel in the input image. The sparse hypercolumn descriptors are then fed to a non-linear classifier, in our case, a 2-layered multi-layer perceptron network (again, implemented as 1x1 convolutions) with 256 neurons respectively. We use a sparsely-sampled output mask, whose pixels correspond to the location of the sparse hypercolumns to learn pixel-wise class predictions as shown in Figure 2. The network was implemented using Keras package [12].

Hypercolumn descriptor can be expressed as in (1)

$$h(p) = [c_1(p), c_2(p), c_5(p)] \tag{1}$$

where h(p) is the hyper column feature for the pixel, $c_i$(p) where $i = 1,2,5$ represents the feature vector corresponding to the ith layer. Given an input, our SDCS network generates pixel-level predictions by operating over the hypercolumns. The final prediction of the pixel p is given by equation (2). We estimate the prediction probability $f_{\theta,p}$(X) for every pixel p as a function of activation given a set of hypercolumn descriptors and its parameter set $\theta$:

$$f_{\theta,p}(X) = g(h_p(X)) \tag{2}$$

### 2.4 Spatially Constrained Convolutional Neural Network (SCCNN)

We used the same training patches of SDCS network to further classify the detected nucleus using SCCNN framework. We augmented the training regions by flip and mirror operations on individual patches. The per pixel prediction centers of individual cells forms the input to the spatially constrained layer. This framework uses the sliding window strategy with overlapping windows. The predicted probability of being center of the nucleus is generated for each patch size of 51x51 from the constrained network. Subsequently, the results are aggregated to form the probability map representing the local maxima at the nuclei centers. An empirical threshold is determined to our dataset to remove any over prediction. All local maxima whose probability is less than the threshold are not considered in our final output detection. The nucleus centers are classified into respective four classes using the integrated SDCS detection and the SCNN framework (Figure. 3).

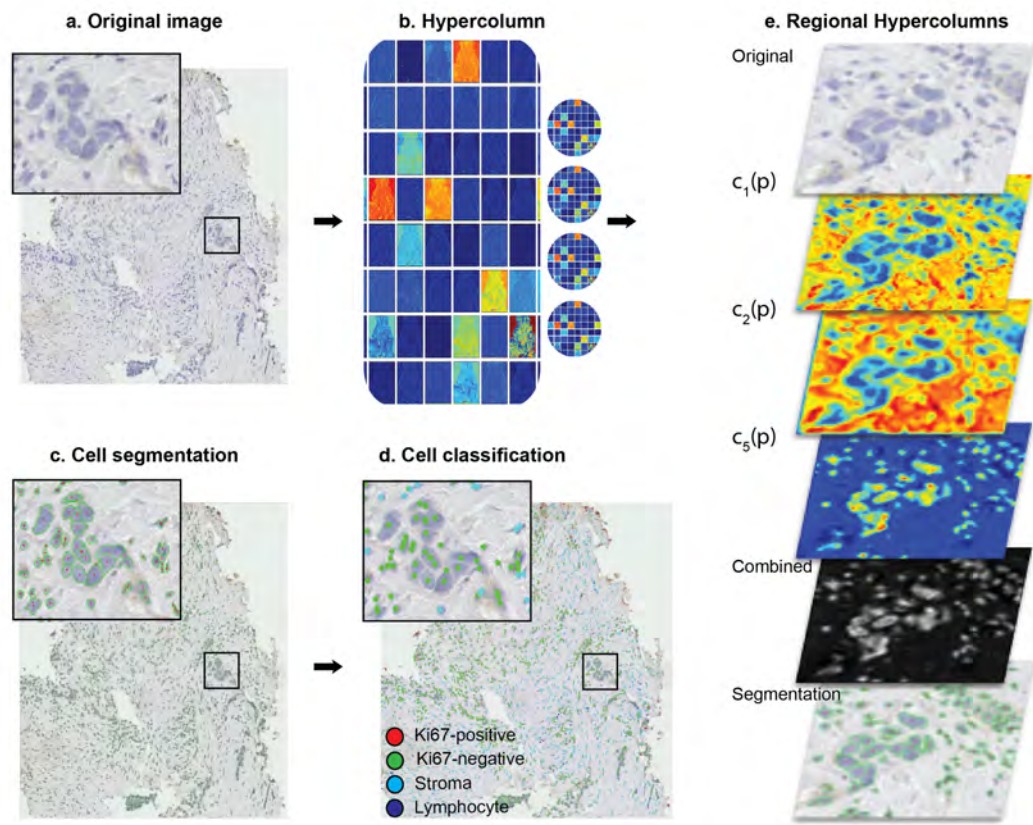

Figure 3: a) Schematic of our simultaneous detection and cell segmentation (SDCS) network hypercolumn generation. b) The exemplary hypercolumn feature representation from the convolution ($c_1$(p)) for the original image to extract the hypercolumn descriptors. c) The hypercolumn descriptors is then fed to the dense multi- layer perceptron network to segment them into Ki67 positive and hematoxylin positive cells. d) Output of the integrated SDCS framework classifying the individual nuclei centers into Ki67 positive, Ki67 negative, stromal cell and lymphocyte respectively. e) Regional hypercolumn descriptor feature maps generated from $c_1$(p), $c_2$(p),$c_5$(p).

## 2.5   SVM classification

Localization of individual cells involves morphological image processing and it includes noise removal using median filter, morphological gradient calculation, otsu threshold and distance transformation followed by watershed segmentation to identify the local maxima. The segmented cell center is used to estimate the features. 101 hand crafted features were estimated from each nucleus, comprising 26 haralick feature [14] extracted for the training images in the gray channel and hematoxylin channel, 49 translational and rotational invariant zernike moment features [15], 11 nuclei based features (mean distance of the nuclei center from the perimeter, standard deviation of the distance of nuclei center to the perimeter, nuclear area, major axis, minor axis, eccentricity, orientation, first Hu moment, second Hu moment, nuclei shape roundedness), 15 intensity based features from individual color channel of the RGB image (mean, standard deviation, variance, skewness, kurtosis). The features were normalized and then used to normalize the test dataset. A support vector machine (SVM) with a radial basis function (RBF) was trained to classify the cells into 4 classes. Results from the classification were further used to evaluate the performance of the model on two independent test sets (Table 2).

# 3 Experimental Results

## 3.1 Quantitative Comparison

We compare cell segmentation and classification results between the integrated SDCS network and machine-learning based SVM method. As demonstrated in our experimental results (Table 2), average classification accuracy increased from 82.05% (SVM) to 99.06% (deep SDCS) on the test set 1 and from 76.5% to 89.59 % on test set 2. Surprisingly, we also noted that a network configuration with the conventional fine tuning of VGG16 with the training regions but without extraction of hypercolumns resulted in a lower accuracy, compared to our integrated SDCS network that uses activation from few convolution layers to construct the hypercolumn based feature maps extracted from $c_1(p)$, $c_2(p)$,$c_5(p)$] (Table 2).

Table 2: Comparison of classification accuracy of machine learning approach based on SVM classification and deep learning approach based on integrated SDCS network classification on the test dataset comprising images from two sets.

| Approach | Average accuracy set1 | Average accuracy set2 |
|---|---|---|
| Haralick Texture | 82.05% | 76.5% |
| SDCS Network without hypercolumns | 85.08% | 82.3% |
| Integrated SDCS Network | 99.06% | 89.59% |

Further breakdown of evaluation metrics for the integrated SDCS network on two test sets indicates better F1-score, precision and recall in test set 1 (Table 3). To evaluate the integrated SDCS framework for the classification of Ki67 images, we use following two test sets. POETIC (1035 cells in 2 slide images from test set 1) and pilot POETIC (1705 in 5 slide images from test set 2).

Table 3: Breakdown of the evaluation metrics on two test sets based on integrated SDCS network classification.

| SDCS Network | Test set 1 | Test set 2 |
|---|---|---|
| Average accuracy | 99.06% | 89.59% |
| Average precision | 99.06% | 82.49% |
| Average Recall | 99.06% | 75.70% |
| Average F1-score | 99.5% | 77.17% |
| Average specificity | 99.7% | 90.20% |
| Average sensitivity | 99.5% | 75.70 % |

## 3.2 Visual Inspection

We show that the classification results (Figure. 4.) based on our integrated SDCS method and the manual method from a tile of individual slide used in our calculation of evaluation metrics on test dataset in Figure.4. For our validation experiment, we used our integrated model to predict class label for known cell locations that had manual annotations. The nuclei in the test dataset set1 were weakly stained and were frequently hollow with non-uniform staining. It can be observed that our integrated SDCS based classification results were closer to the manual annotations than classical SVM based method (more obvious in the zoomed views of the region in Figure. 4). The model has learnt the complex representations of the Ki67 -negative nuclei and positive nuclei.

# 4 Discussion

In this work we propose an integrated deep SDCS framework for simultaneous detection and classification of multiple cells in Ki67-stained IHC images. A major challenge in analyzing these images is the heterogeneous staining, which often results in under detection as we demonstrated with classical machine learning methods. To detect weakly stained cells without resulting in over-detection, hypercolumn descriptors were used to integrate activations from multiple convolutional layers, capturing image semantics across granularities. Another advantage for this network is that only annotations for cell centroids are needed. Therefore,

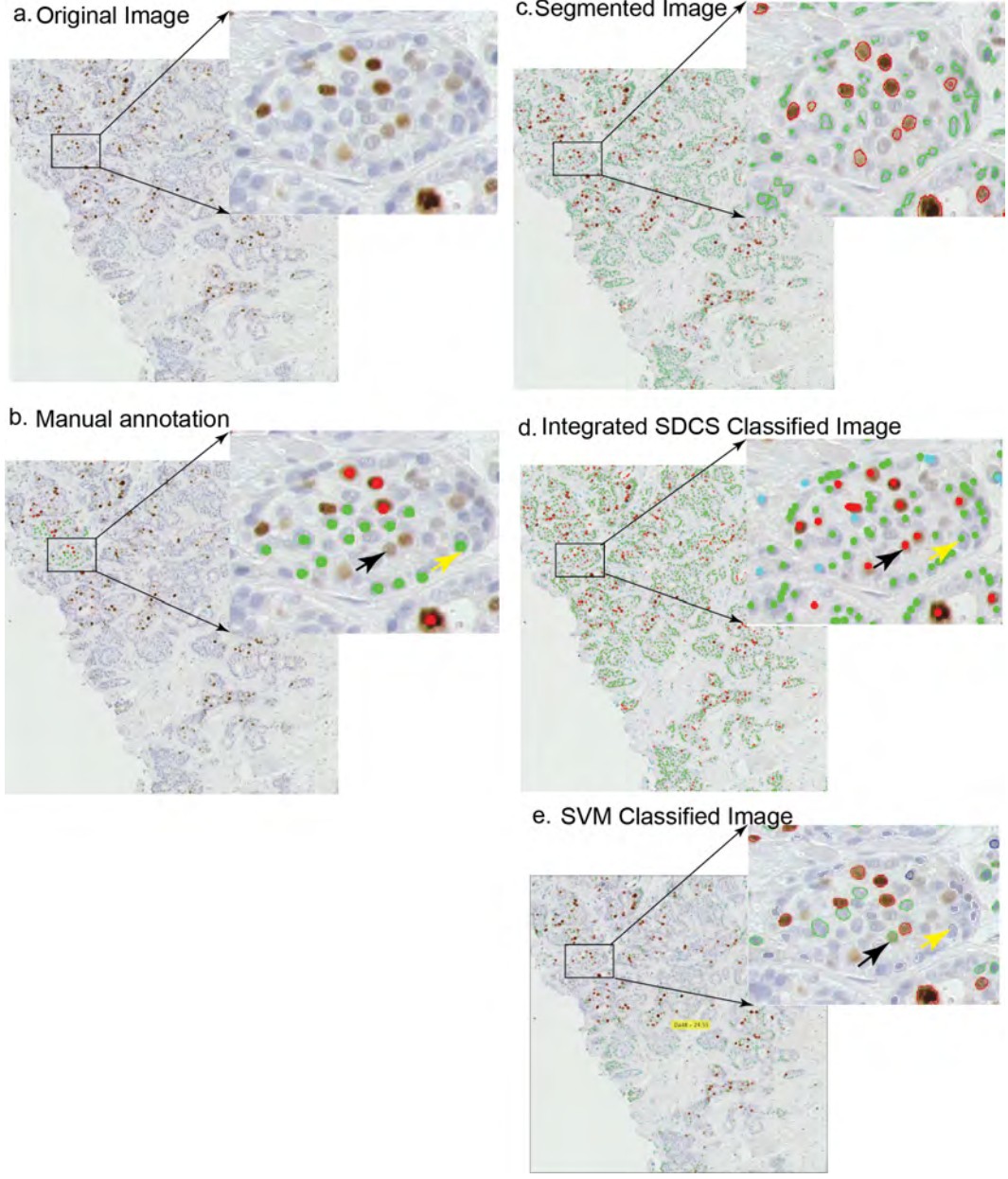

Figure 4: Visual comparison of classification based on weakly stained region of biopsy indicating the classification based on integrated SDCS network and SVM classification. a) original image tile (2000x2000) b) Manual annotated tile c) SDCS detection and segmentation d) Integrated SDCS classification e) SVM classification. Black arrow indicates the misclassification error between the two approaches. Yellow arrow indicates the under segmentation error due to the morphological processing. Integrated SDCS detection and classification closer to manual annotation.

compared with existing networks such as U-net [16, 17], our method removes the need for laborious cell area annotations, which can dramatically speed up the annotation collection and training processes. Moreover, our recognition framework integrates problem-specific design such as a tissue region mask to remove coverslip artefact, which is not part of other standard networks.

The architecture based on hypercolumns was empirically chosen based on our evaluation of the training data. The integrated SDCS approach localizes features from multi-scale RGB patches, and extract them from the activation maps to construct the hypercolumn descriptors. The importance of such hypercolumn descriptors was reflected in our experimental results comparing SDCS network with and without hypercolumn. Detection was followed by non-linear classification by the SCCNN method. Overall, we found that this pipeline was able to overcome the challenge in detecting weakly stained Ki67-negative nuclei. No empirical threshold was needed in our segmentation and detection framework.

In comparison, the texture-based approach uses features extracted from grey and hematoxylin channels. Visual inspection of the results from the svm classification approach indicated that hand-crafted feature-based model output detecting fewer Ki67 –negative cells compared to the manual annotation. Firstly, this is due to weakly stained samples, where the segmentation needs further refinement and classification output cannot overcome this under segmentation. Secondly, we observed misclassifications of Ki67 -positive cell as Ki67 -negative cell in svm classification than the SDCS approach. (Figure. 4). Thus, the hand-crafted feature extracted from single resolution lacks information to discern weakly stained nuclei as opposed to the multi-scale feature extraction in the integrated SDCS method. Additionally, empirical thresholds for otsu thresholding and distance transform were necessary for obtaining the cell segmentation. Bankhead, P. et al. [9] used QuPath, a texture based method based on classical machine learning, to analyse tissue micro arrays. This method is yet to be analytically validated for biopsy and whole slide images. We were unable to directly compare with QuPath, however, due to the difficulty in parsing our training data.

Direct output of the proposed framework is the single-cell classification of stroma, lymphocytes and Ki67-positive and negative cancer cells. Limitations include the relatively small training and test sets, which we plan to expand on by taking more images from the POETIC study consisting of a total of 9,000 Ki67 slides. In addition, direct comparison of our method with the other state-of-the-art networks for classification was not possible, because often the entire image patch is classified. Because our training data are limited to cell centroids instead of cell areas, we cannot directly evaluate cell segmentation accuracy.

In summary, visual assessment is considered as the gold standard for Ki67 scoring. Inter- and intra- observer bias may, however, affect the results. Automated estimation of Ki67 score using our integrated SDCS method has the potential to provide consistent and reproducible scores, following training by expert pathologists. Our method also classifies lymphocytes and stromal cells, which could be further used in clinical studies to better understand the complex tumor microenviroment.

## 5  Conclusion

We have demonstrated the performance of our new pipeline for simultaneous cell detection and classification in samples from a large breast cancer clinical trial. Specifically, its use of hypercolumn descriptors aids cell segmentation and further classification in heterogeneously stained Ki67 immuno-histochemistry images. Compared with a classical machine learning method, this approach detects weakly stained nucleus with higher precision. This integrated framework will allow us to further validate and test automated Ki67 scores in large breast cancer patient cohorts.

## 6  Acknowledgments

This work is supported by Breast Cancer Now (2015NovPR638). Also the authors would like to thank Prof Nasir Rajpoot at Tissue Image Analytics Lab, University of Warwick for their help in implementation of SCCNN [11].

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
