# OpenReview forum: "DeepSDCS: Dissecting cancer proliferation heterogeneity in Ki67 digital whole slide images"
_MIDL.amsterdam/2018/Conference — MIDL 2018 Poster_

### Review · AnonReviewer1 · 2018-05-09
**This paper presents a novel cell detection and classification method in Ki67 whole slide images. They have proposed the simultaneous detection and cell segmentation method based on the features from a hypercolumn layer. The experiment results have been shown that the proposed method has the better performance than other conventional methods.**

**Rating:** 4
**Confidence:** 3

**Review:**


Quality & Clarity

#1. This paper is well organized, especially discussion section.
#2. Also, the description of dataset and experimental results is well written.
#3. Author have well defined the problem to be solved.

Originality & Significance

(+) This paper presents a novel cell detection and segmentation method by integrating a hypercolumn layer to spatially constrained convolutional neural networks.
(+) In order to prove the novelty of the proposed system, they have well designed the experiments. The method with a hypercolumn layer would be very helpful in improving performance.
(+) The proposed method is more robust to staining quality than other methods.
(-) Why does the performance of two test sets differ significantly?
(-) Author didn’t report the performance based on the other architecture that has more powerful representation (e.g., ResNet, Inception, etc).
(-) Author should compare with other simultaneous framework such as Mask R-CNN.
(-) How do they define the cells in background region? How did they handle with cells that did not belong to the four classes?


**Special Issue:**

No

---

### Review · AnonReviewer3 · 2018-05-10
**Hypercolumns for improved detection and classification of cells in Ki67 whole slide images**

**Rating:** 2
**Confidence:** 1

**Review:**

Clarity:
+ The paper is well-written and the problem is clearly motivated.
- It is difficult at times for the layman to get a clear picture of the whole framework, particularly how the SDCS network and SCCNN framework are integrated. Which part of the framework is responsible for detection/localisation/classification and how they interact with each other could maybe be conveyed in a simpler way, so that the reader does not have to backtrack as much to the original SCCNN work.

Contribution:
+ The authors experiment with sparse hypercolumn descriptors for improved localisation of nuclei centers and classification. They show improved accuracy on the classification task compared to hand-crafted features + SVM.
- Hypercolumns for object detection and segmentation were proposed in [12]. While they may not have been as extensively studied in the community as e.g., U-Nets, they follow a similar vein. U-Nets and other richer architectures using skip/residual connections are potentially more powerful, and would be natural frameworks with which to compare the proposed method. Comparison with the standard SCCNN approach would also make sense.

Comments:
* The SCCNN framework proceeds in two stages: an SC-CNN for nucleus detection, then a softmax CNN on patches centered at detected locations for classification. It is somewhat unclear where the authors depart from that framework. Are hypercolumn features fed to both the spatially constrained layers and the classification network?
* Is the architecture trained end-to-end? It would be very helpful for the reader to see the cost function that is optimised.
* The terms segmentation, detection and classification are used interchangeably at times in the text and in figure captions. Just to clarify, is voxel-wise segmentation, in contrast to nucleus center localisation, performed at any point in the pipeline or is it indeed entirely bypassed?

* Are the VGG network weights pretrained or is the nextwork trained from scratch?
* How and at which stage is the selection of hypercolumns (from dense to sparse) performed?

* What explains the gap in performance between the two test datasets?
* Is there a way to isolate the respective gains in terms of nucleus localisation and classification with the proposed approach vs. alternative approaches?

**Special Issue:**

No

---

### Review · AnonReviewer2 · 2018-05-17

**Rating:** 5
**Confidence:** 3

**Review:**

Ki67 IHC images is the basis for estimating the proliferation status of breast cancer. Cells are simultaneously detected and characterised as being on of four classes. The method clearly outperformed standard approaches such as using Haralick features and SVM.

**Special Issue:**

No

---

### Decision · Program_Chairs · 2018-05-15
**Paper84 Acceptance Decision**

Poster